

# Winter brown carbon over six China's megacities: Light
# absorption, molecular characterization, and improved
# source apportionment revealed by multilayer perceptron
# neural network
Diwei Wang[1], Zhenxing Shen[1*], Qian Zhang[2], Yali Lei[3], Tian Zhang[1], Shasha Huang[1],
Jian Sun[1], Hongmei Xu[1], Junji Cao[4]
[1]Department of Environmental Science and Engineering, Xi'an Jiaotong University, Xi'an 710049,
China
[2]Key Laboratory of Northwest Resource, Environment and Ecology, MOE, Xi'an University of
Architecture and Technology, Xi'an 710055, China
[3]Key Lab of Geographic Information Science of the Ministry of Education, School of Geographic
Sciences, East China Normal University, Shanghai 200241, China
[4]Key Lab of Aerosol Chemistry & Physics, SKLLQG, Institute of Earth Environment, Chinese Academy
of Sciences, Xi'an, China
*Correspondence to:* Zhenxing Shen (zxshen@mail.xjtu.edu.cn)
**Abstract.** Brown carbon (BrC) constitutes a large fraction of organic carbon and exhibits strong light
absorption properties, thus affecting the global radiation budget. In this study, we investigated the light
absorption properties, chemical functional bonds, and sources of BrC in six megacities in China, namely
Beijing, Harbin, Xi'an, Chengdu, Guangzhou, and Wuhan. The average values of the BrC light
absorption coefficient and the mass absorption efficiency at 365 nm in northern cities were higher than
those in southern cities by 2.5 and 1.8 times, respectively, demonstrating the occurrence of abundance of
BrC in northern China's megacities. Fourier transform–infrared (FT-IR) spectra revealed sharp and
intense peaks at 1640, 1458–1385, and 1090–1030 $cm^{-1}$, which were ascribed to aromatic phenols,
confirming the contribution of primary emission sources (e.g., biomass burning and coal combustion) to
BrC. In addition, we noted peaks at 860, 1280–1260, and 1640 $cm^{-1}$, which were attributed to
organonitrate and oxygenated phenolic groups, indicating that secondary BrC also existed in six
megacities. Positive matrix factorization (PMF) coupled with multilayer perceptron (MLP) neural
network analysis were used to apportion the sources of BrC light absorption. The results showed that
primary emissions (e.g., biomass burning, tailpipe emissions, and coal combustion) made a major
contribution to BrC in six megacities. However, secondary formation processes made a greater



contribution to light absorption in the southern cities (17.9%–21.2%) than in the northern cities (2.1%–
10.2%). These results can provide a basis for the more effective control of BrC to reduce its impacts on
regional climates and human health.
**1 Introduction**
Brown carbon (BrC) constitutes a vital fraction of carbonaceous aerosols and exhibits strong light
absorption properties in near-ultraviolet (UV) and visible wavelength regions (Laskin et al., 2015; Wu et
al., 2021; Zhang et al., 2022). Therefore, it has received extensive attention in recent years (Laskin et al.,
2015; Yan et al., 2018; Yuan et al., 2020). BrC has substantial effects on radiative forcing, cloud
condensation, ice cores, and climate (Ma et al., 2020; Sreekanth et al., 2007). On the basis of remote
sensing observations and chemical transport model results, studies have detected a BrC-induced
nonnegligible positive radiative forcing ranging from 0.1 to 0.6 W m$^{-2}$ on a global scale (Jo et al., 2016;
Wu et al., 2020).
BrC in urban atmospheres can originate from numerous sources, including incomplete combustion of
fossil fuels, biomass burning, forest fires, and residential coal combustion (Kirchstetter et al., 2004; Shen
et al., 2017; Soleimanian et al., 2020). In addition, both primary BrC and gaseous pollutants emitted from
anthropogenic and biological activities can be converted into secondary BrC through a series of
atmospheric chemical reactions (Kumar et al., 2018; Laskin et al., 2015). Studies have determined that
the absorption properties of BrC exhibited distinct temporal and spatial variations in different regions
and cities, and these properties were closely related to diverse emissions sources and complex
atmospheric aging processes (Chung et al., 2012; Wu et al., 2021). For example, Devi et al (2016)
observed that BrC contributed differently to light absorption in the rural and urban southeast United
States. Furthermore, a stronger light absorption ability in cold seasons (fall and winter) in Beijing, Xi'an,
Taiyuan, Seoul, and other cities has been found to be strongly associated with increased biomass burning
emissions (Cheng et al., 2016; Kim et al., 2016; Mo et al., 2021; Shen et al., 2017). Another study noted
that secondary organic aerosol (SOA) formation processes constituted a major source of BrC in Atlanta
and Los Angeles; moreover, the optical properties of BrC differed considerably between the two cities
due to differences in secondary BrC precursors (Zhang et al., 2011).
China has a high concentration of atmospheric water-soluble organic carbon, which has a major impact



on regional air quality, visibility, and the climate (Mo et al., 2021). However, to our knowledge, limited
study was conducted to insight to the optical profiles, molecular composition, and sources apportionment
of BrC in a large scale in China. Accurately understanding the spatial variations of the sources and light
absorption properties of BrC in China is essential for reducing uncertainty about the effects of BrC on
the climate. Many studies have used receptor modelling techniques such as positive matrix factorization
(PMF) coupled with multiple linear regression analysis to assign the sources of BrC (Bao et al., 2022;
Lei et al., 2019; Soleimanian et al., 2020). However, atmospheric processes are generally non-linear in
nature, thus traditional deterministic models could be limited. The artificial neural network (ANN) based
models, such as multilayer perceptron (MLP), have been shown to provide meaningful results closer to
realistic estimates than most linear models (Borlaza et al., 2021; Elangasinghe et al., 2014). Therefore,
in this study, a winter campaign for PM$_{2.5}$ sampling was conducted over six China's megacities. The
purposes of this study were to 1) investigate the spatial variations of the carbonaceous matter
concentrations and optical properties of BrC across six representative urban areas in China, 2) determine
the molecular composition of BrC, and 3) insight the relationship between light absorption and BrC
sources by using PMF coupled with ANN-MLP.

## 2 Methods

### 2.1 Samples collection

PM$_{2.5}$ samples were collected in six cities in China (Figure 1): three cities in northern China (Beijing
[BJ], Harbin [HrB], and Xi'an [XA]) and three cities in southern China (Chengdu [CD], Guangzhou
[GZ], and Wuhan [WH]). We classified the cities as being in northern or southern cities according to their
geographic location, such as "north or south of the Huaihe River". Owing to geographical factors, these
cities exhibit considerable differences in terms of energy structure and climate. The average annual
temperature in northern cities is generally below 15°C, while in southern cities it is usually above 15°C
(Mo et al., 2021). Information about the six cities and the sampling sites is summarized in Table S1
(Supporting Information).



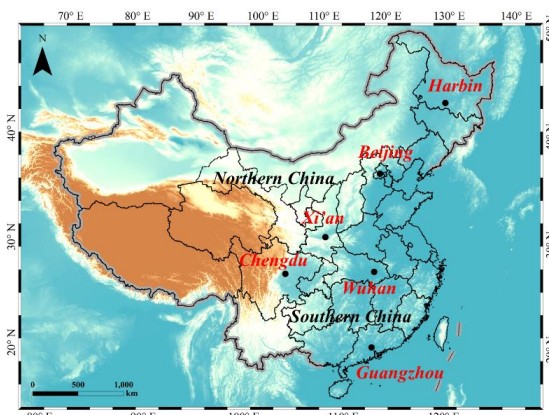


**Figure 1. PM$_{2.5}$ samples were taken in six Chinese cities.**

For sample collection, filter samplers were mounted on rooftops between 8 and 30 m above the ground,
and samples were collected from November 20 to December 22, 2019. In BJ, HrB, and GZ, a mini-
volume sampler operating at 5 L min$^{-1}$ (Airmetrics, Springfield, OR, USA) was used to collect PM$_{2.5}$
samples on 47-mm quartz-fiber filters (Whatman, Maidstone, UK) for 24 h. In CD, a medium-volume
PM$_{2.5}$ sampler operating at 100 L min$^{-1}$ (HY-100SFB, Hengyuan, Qingdao, China) was used to collect
PM$_{2.5}$ samples on 90-mm quartz-fiber filters (Whatman). Moreover, in XA and WH, a high-volume
sampler (HVS-PM$_{2.5}$, Thermo-Anderson Inc. Cleves, OH, USA) with a flow rate of 1.13 m$^3$ min$^{-1}$ was
used to collect PM$_{2.5}$ samples on quartz-fiber filters (203 mm × 254 mm, Whatman, QMA). Before
sample collection, all quartz filters were prebaked at 780 °C for 7 h to eliminate any residual carbon. A
detailed description of the quality control procedures for the filters before and after the sampling
processes can be found in the article by Shen et al (2017). After the sampling processes, the samples were
sealed and stored below 0 °C to avoid evaporative losses before analysis.
**2.2 Chemical analysis**
The organic carbon (OC) and elemental carbon (EC) of the PM$_{2.5}$ samples were analyzed using a
Thermal and Optical Carbon Analyzer (DRI Model 2001A, Atmoslytic, Inc., USA) in accordance with
the improved Interagency Monitoring of Protected Visual Environment (IMPROVE) thermal/optical
reflectance protocol. Detailed descriptions of the OC and EC measurement methods can be found in the
article by Cao et al (2004). A portion of each filter (about 2.84 cm$^2$) was extracted using 10 mL of
ultrapure water to analyze water-soluble inorganic ions (Na$^+$, NH$_4^+$, K$^+$, Mg$^{2+}$, Ca$^{2+}$, Cl$^-$, NO$_3^-$, and SO$_4^{2-}$)





through ion chromatography (Dionex 500, Dionex Corp, USA). A detailed description of the ion analysis
method used in this study can be found in the article by Shen et al (2008).

**2.3 Optical properties of methanol extracts**

A 0.526-cm$^2$ punch was ultrasonically extracted from each filter sample by using 5 mL of methanol
(HPLC Grade, Fisher Scientific, NH, USA) for 30 min. Subsequently, all extracts were filtered through
a microporous membrane with a diameter of 25 mm and pore size of 0.22 μm (Puradisc 25 TF, PTFE
membrane) to remove insoluble components. The UV–visible absorption spectra of the BrC samples
were determined using a liquid waveguide capillary cell–total OC spectrophotometer (LWCC-2100,
World Precision, Sarasota, FL, USA) between the wavelengths of 200 and 700 nm. The BrC optical
properties such as $b_{abs365,\ methanol}$ (The absorption coefficient for methanol exacts at 365 nm) and $MAE_{365,}$
$_{methanol}$ (normalized by $b_{abs365,\ methanol}$ to organic carbon, OC) were calculated as showed in previous study
(Lei et al., 2019) and details was listed in Text S1.

**2.4 Fourier transform infrared spectroscopy spectra**

Functional groups in the samples collected in six megacities were characterized using a Fourier
transform infrared (FT-IR) spectrometer (Bruker Optics, Billerica, MA, USA). The method described in
section 2.3 was used to extract the BrC filtrates, then the BrC extracts were concentrated to 0.5 mL under
a gentle nitrogen flow, after which they were mixed with 0.2 g of KBr (FT-IR grade, Sigma-Aldrich) and
then blown with nitrogen to complete dryness. The resulting extract–potassium bromide mixture was
ground in an agate mortar and examined through FT-IR spectroscopy. The FT-IR spectrum of each sample
was recorded in transmission mode by averaging 64 scans using a standard optical system with KBr
windows. The spectra were recorded in the wavelength range of 4000–400 cm$^{-1}$ at a resolution of 4 cm$^{-1}$.
Before analyzing the aerosol extract samples, we obtained the baseline spectrum by analyzing pure KBr.

**2.5 Source apportionment of BrC light absorption coefficient at 365 nm**

In this study, the source apportionment of BrC was conducted using the PMF coupled with ANN-MLP
methods by following the steps: 1) identification and quantification of the major sources of PM$_{2.5}$ for the
six cities using PMF (The United States Environmental Protection Agency, PMF 5.0); 2) produces a
predictive model by ANN-MLP for one variable (BrC b$_{abs365}$) based on the values of the input variables



(PM$_{2.5}$ sources daily contributions). PMF is a bilinear factor model that has been widely used in source
apportionment studies (Shen et al., 2010; Cao et al., 2012; Lei et al., 2018; Li et al., 2021; Tao et al.,
2017). In the present study, water-soluble inorganic ions (Na$^+$, NH$_4^+$, K$^+$, Mg$^{2+}$, Ca$^{2+}$, NO$_3^-$, SO$_4^{2-}$ and
Cl$^-$) and carbon fractions (OC1, OC2, OC3, OC4, EC1, and EC2) were used as data inputs for PMF. The
PMF model was run multiple times, extracting four to six factors. A more detailed description of these
items can be found in the article by Lei et al (2019). Subsequently, an MLP model was constructed. The
model was developed using IBM SPSS Statistics for Windows, version 23 (IBM Corp., Armonk, NY,
USA). The detail information of the ANN-MLP model construction and training was described in Text
S2. After ANN-MLP model training, the obtained MLP model was applied to a set of virtual datasets.
Each virtual dataset consists of each source with the same mass contribution (from PMF analysis) as the
original dataset, but with one source set to zero. The b$_{abs365}$ contribution for a specific source was obtained
by subtracting the b$_{abs365}$ simulation value obtained using the virtual dataset from the b$_{abs365}$ simulation
value obtained using the original MLP model, which contains all the source contributions (Borlaza et al.,

2021).

**3 Results and discussion**
**3.1 General description of PM$_{2.5}$ and its chemical species in six megacities**

As presented in Table S2, the PM$_{2.5}$ concentrations in six cities ranged from 9.9 to 241.9 μg/m$^3$ and

exhibited a significant spatial variation ($p < 0.01$), indicating the complexity of air pollution and spatial
differences in air pollution levels in China. HrB had the highest average PM$_{2.5}$ concentration ($85.5 \pm 43.9$
μg/m$^3$), which exceeded National Air Quality Standard grade-II (24-h average: 75 μg/m$^3$) and was 1.5,
1.1, 1.2, 2.0 and 1.3 times higher than those recorded in BJ, XA, CD, GZ, and WH, respectively. This
phenomenon indicates that PM$_{2.5}$ pollution is still a major challenge in China, particularly in northern
China.

The average concentration of OC, a major chemical component of PM$_{2.5}$, ranged from 5.6 to 19.4

μg/m$^3$ in six megacities; these cities can be arranged (in descending order) as follows in terms of the
average OC concentration: HrB > XA > BJ > WH > GZ > CD (Table S2). Similar to the trend observed
for PM$_{2.5}$, the average OC concentration in the northern cities ($15.5 \pm 7.9$ μg/m$^3$) was higher than that in
the southern cities ($9.2 \pm 4.6$ μg/m$^3$); this can primarily be attributed to substantial emissions from





residential heating in winter in northern China (Lei et al., 2018; Sun et al., 2017; Zhang et al., 2021). To
assess the sources of atmospheric BrC, we estimated the concentrations of primary OC (POC) and
secondary OC (SOC) by using the EC tracer method (Ram and Sarin, 2011). As presented in Table S2,
the average SOC concentrations throughout the measurement period ranged from 1.0 (CD) to 9.2 $\mu g/m^3$
(HrB), and the fractional contributions of SOC to OC varied from 22.6% to 66.6%. The average POC
concentrations ranged from 4.0 (GZ) to 10.2 $\mu g/m^3$ (HrB), and POC constituted 34.4%–77.4% of the
total OC mass in the six cities. Accordingly, the SOC and POC concentrations exhibited typical spatial
fluctuations, which were consistent with the fluctuations of the $PM_{2.5}$ and total OC concentrations. These
results reveal that primary emissions usually dominated secondary formation processes, especially in the
northern cities.

**3.2 Light absorption properties of BrC**

As plotted in Figure 2, the light absorption coefficient ($b_{abs}$, $Mm^{-1}$) values for BrC exhibited significant
spatial variations across the six cities (1.7–64.1 $Mm^{-1}$; $p < 0.01$). We executed Student $t$ test at the 95%
confidence level and observed that HrB had the highest average $b_{abs365}$ value (29.3 ± 14.2 $Mm^{-1}$),
followed by BJ (11.4 ± 3.9 $Mm^{-1}$), WH (10.0 ± 3.2 $Mm^{-1}$), XA (8.3 ± 2.4 $Mm^{-1}$), CD (5.6 ± 2.7 $Mm^{-1}$),
and GZ (4.3 ± 1.4 $Mm^{-1}$). The average $b_{abs365}$ value in the northern cities was 15.7 ± 12.3 $Mm^{-1}$, which
was 2.5 times higher than that in the southern cities ($p < 0.01$). The large variation in the measured $b_{abs365}$
values in these megacities was observed, which reflected that the light absorption of BrC was heavily
affected by chromophore sources (Huang et al., 2018; Soleimanian et al., 2020), aging during
atmospheric transportation (Lambe et al., 2013), and meteorological conditions (Li et al., 2021). Light-
absorbing carbonaceous aerosols were believed to be responsible for the considerable absorption of light
in the atmosphere (Xie et al., 2020). As presented in Figure S2, we observed positive correlations between
$b_{abs365}$ and POC in the six cities ($r$ range: 0.61–0.92). Similar correlations were observed between $b_{abs365}$
and SOC ($r$ range: 0.51–0.80), indicating that the sources of atmospheric BrC in the six cities were quite
complex. Apart from primary emissions, secondary formation processes also seemed to have a
considerable contribution to BrC in these cities. The $b_{abs365}$ values in HrB, BJ, XA, and WH are within
the range of values observed previously in Beijing (4–75 $Mm^{-1}$; Cheng et al., 2016) and the Indo-
Gangetic Plain (3–457 $Mm^{-1}$; Satish et al., 2020). Biomass burning was revealed to be the dominant
source of BrC in these cities during winter (Elser et al., 2016; Shen et al., 2009; Sun et al., 2017).



Furthermore, we observed high correlations ($r$ range: 0.69–0.92) between $b_{abs365}$ and $K^+$, which is
commonly regarded as a tracer of biomass burning (Shen et al., 2010), in HrB, BJ, XA, and WH (Figure
S3). This evidence supports the aforementioned findings that emissions from biomass burning might be
the major BrC source in winter in these cities. For the southern cities CD and GZ, the low $b_{abs365}$ values
(1.7–11.5 $Mm^{-1}$) are of the same order of magnitude as those reported previously in Nanjing (3.3–13
$Mm^{-1}$; Chen et al., 2019; Chen et al., 2018), Seoul (0.9–7.3 $Mm^{-1}$; Kim et al., 2016), and Hong Kong
(4.8–10.6 $Mm^{-1}$; Zhang et al., 2020). The aging or oxidation of aerosols was confirmed to be the major
source of BrC in these regions, indicating that secondary aerosols are likely a major source of winter BrC
in CD and GZ.

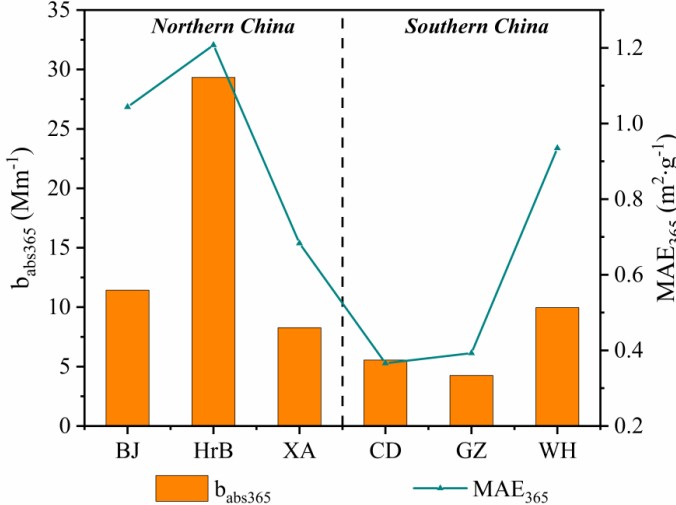


**Figure 2. Spatial variations of BrC light absorption properties from six Chinese cities. The bars represent the**
**light absorption coefficient at 365 nm ($b_{abs365}$, left axis), and the lines represent the mass absorption efficiency**
**at 365 nm ($MAE_{365}$, right axis).**
The mass absorption efficiency (MAE, $m^2 g^{-1}$) is a key parameter for describing the light absorption
ability of atmospheric BrC (Li et al., 2021; Peng et al., 2020). Figure 2 illustrated the average MAE
values measured at 365 nm ($MAE_{365}$) in the six cities; compared with the value measured in CD (0.37 ±
0.18 $m^2 g^{-1}$), those measured in the other five cities were higher by 1.1–3.3 times. These cities can be
arranged as follows (in descending order) in terms of the measured $MAE_{365}$ values: HrB > BJ > WH >
XA > GZ > CD. These differences in $MAE_{365}$ values can be attributed to the variance of the light



absorption capacity of BrC in different megacities. The average $MAE_{365}$ values measured for BrC in BJ,
HrB, XA, and WH (range: 0.68–1.21 $m^2 g^{-1}$) were within the MAE ranges of biomass burning, such as,
the average $MAE_{365}$ measured for BrC were $0.97 \pm 0.26$ $m^2 g^{-1}$ for wood burning (Du et al., 2014), 1.05
$\pm 0.08$ $m^2 g^{-1}$ for corn stalk combustion (Du et al., 2014), and $1.28 \pm 0.12$ $m^2 g^{-1}$ for wheat stubble burning
(Xie et al., 2017; Lei et al., 2018), indicating that biomass burning may be a major source of winter BrC
in these cities. Biomass burning is commonly regarded as the main emission source for BrC, which has
a high absorption capacity, as indicated by field observations and model predictions (Desyaterik et al.,
2013; Feng et al., 2013; Lei et al., 2018). Notably, the $MAE_{365}$ values derived for BrC emitted from
primary fossil fuel combustion are similar to those derived for biomass burning (Yan et al., 2017); for
example, former studies have revealed that the $MAE_{365}$ values for BrC produced by primary emissions
from residential coal combustion were in the range of 0.30–1.51 $m^2 g^{-1}$ (Ni et al., 2021; Yan et al., 2017).
Therefore, coal combustion may also be a potential source of BrC in these cities. By contrast, we
observed lower average $MAE_{365}$ values for BrC in GZ and CD (range: 0.37–0.39 $m^2 g^{-1}$). Previous studies
have revealed relatively low MAE values for BrC from motor vehicle emissions, including gasoline
vehicle emissions ($0.62 \pm 0.76$ $m^2 g^{-1}$; Xie et al., 2017) and motorcycle emissions ($0.20 \pm 0.08$ $m^2 g^{-1}$;
Du et al., 2014). These findings suggest that the BrC sampled in GZ and CD mainly originated from
traffic emissions. In addition, laboratory experiments in a previous study revealed that $MAE_{365}$ values
decreased from 1.43 to 0.11 $m^2 g^{-1}$ with aerosol aging, which suggests the production of SOA (Ni et al.,
2021). This finding demonstrates that secondary formation processes are among the main sources of BrC
in CD and GZ.

The absorption Ångström exponent (AAE) measurements at 330–550 nm represents the wavelength

dependence of light absorption by BrC (Cheng et al., 2017). We observed that the average AAE values
for BrC varied from 5.4 to 6.8 in the six cities (Figure 3). In general, the AAE values obtained in this
study are higher than those obtained at the Nepal Climate Observatory-Pyramid (3.7–4.0; Kirillova et al.,
2016) and in the Los Angeles Basin ($4.82 \pm 0.49$; Zhang et al., 2013) and lower than those obtained at
the Tibetan Plateau ($8.2 \pm 1.4$; Zhu et al., 2018). Nevertheless, the values obtained in this study are
comparable to those obtained in Beijing (5.3–7.3; Cheng et al., 2016; Wu et al., 2021), Nanjing (6.7;
Chen et al., 2018), the Indo-Gangetic Plain (5.3; Srinivas et al., 2016), New Delhi (5.1; Kirillova et al.,
2014), Seoul (5.5–5.8; Kim et al., 2016), and Xi'an (5.3–6.1; Huang et al., 2018). These similarities can
primarily be attributed to the consistent solubility of chromophores, which are sensitive to the type of

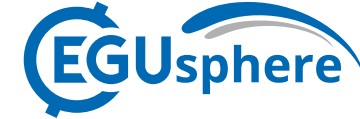



fuel used, the combustion conditions, and the solvents used (Cao et al., 2021; Huo et al., 2018).
Furthermore, the AAE values obtained in this study are within the range of those reported by previous
studies for coal combustion (5.5–6.4; Ni et al., 2021), biomass burning (4.4–8.7; Xie et al., 2017), and
gasoline vehicle emissions (6.2–6.9; Xie et al., 2017). This suggests that BrC in our study may have
multiple sources. Additionally, in contrast to the trends observed for the $b_{abs365}$ and $MAE_{365}$ values for
BrC in the various cities, the AAE values observed in CD and GZ were higher than those observed in the
other cities. A previous study reported that the AAE values for SOA were higher than those for primary
organic aerosols (Saleh et al., 2013), and previous laboratory combustion experiments revealed that the
aging of biomass burning aerosols generally engenders an increase in AAE values (from 6.93 to 15.59;
Sengupta et al., 2018). These findings suggest that BrC in the cities in this study was also affected by
secondary formation processes.

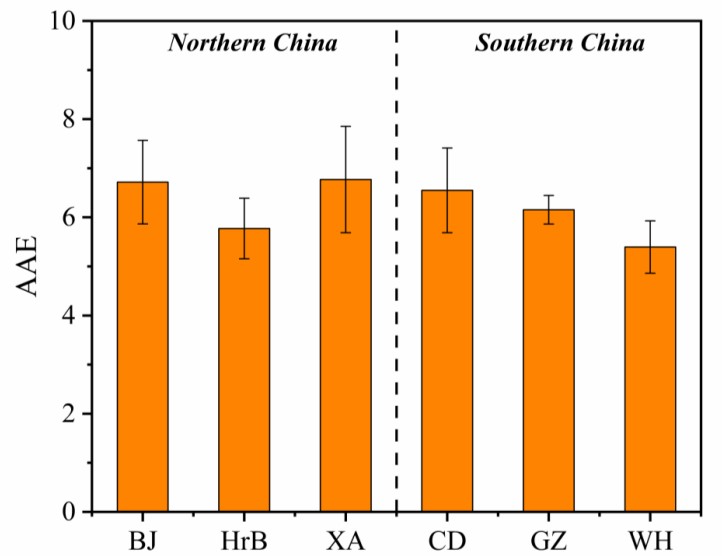


**Figure 3. AAE values of BrC in six cities. AAE is calculated between 330 and 550 nm.**
**3.3 Molecular structure of BrC**
In order to further explore the reasons for the differences in the optical properties of BrC among these
cities, the functional groups of BrC were measured using FT-IR spectroscopy. Figure 4 illustrates the FT-
IR spectra of BrC fractions within the region of 4000–400 cm$^{-1}$ in the six cities. The band in the region
of 400–800 cm$^{-1}$ resulted from the interference from water vapor inside the instrument and thus can be



ignored (Zhang et al., 2020). The broad and strong peak at 3450 cm$^{-1}$ was contributed to the O-H stretch
of H-bonded hydroxyl groups, phenols and carboxylic (Fan et al., 2016; Mukherjee et al., 2020). The
sharp band near 1740 cm$^{-1}$ was usually assigned to the C=O bonds of ketones, quinones, and amides
(Duarte et al., 2005; Kristensen et al., 2015). We also attributed the sharp and intense absorption peaks
at 2850−2990 cm$^{-1}$ to aliphatic asymmetric and symmetric C–H stretching vibrations (Coury and Dillner,
2008). Some bands were also displayed near 1385, 1458 and 1640 cm$^{-1}$, indicating the presence of
aromatic groups (Fan et al., 2016; Zhao et al., 2022). These results demonstrate the complexity of the
chemical composition of BrC in the six cities, mainly containing aliphatic chains, carboxylic groups, and
aromatic groups.

In contrast to these similar functional groups, the apparent differences of typical functional bands were

also found among these cities. The strong band near 3130 cm$^{-1}$ denoting O–H band (Fan et al., 2016;
Mukherjee et al., 2020) were only detected in XA, CD and WH, and the same peak were observed in the
spectra from the corn straw burning (Fan et al., 2016) and coal combustion (Zhang et al., 2022), which
stressed the emissions of biomass burning and coal combustion with high abundance of oxygenated
phenolic compounds in these cities. Moreover, it was noted that the peaks at 1640, 1458, 1385 and 1030
cm$^{-1}$ was significantly higher in HrB, XA and WH than those in other cities. Previous studies confirmed
that these bands were generally ascribed to the C=C and C–H stretching of aromatic rings, O–H bond
deformation and C–O stretching of phenolic groups (Fan et al., 2016; Mukherjee et al., 2020). These
observations indicated the contribution of biomass burning to BrC in winter; this was because that
biomass burning can release heat-modified lignin derivatives such as aromatic phenols (e.g., syringyl
and guaiacyl) (Duarte et al., 2007; Fan et al., 2016; Zhao et al., 2022). Previous studies have shown that
BrC from biomass burning has a high light absorption capacity (Cao et al., 2021; Desyaterik et al., 2013;
Kumar et al., 2018), which supported that these cities with higher abundance of aromatic phenol
functional groups were consisted with higher $b_{abs365}$ (range: 8.3–29.3 Mm$^{-1}$) and MAE$_{365}$ (range: 0.68–
1.21 m$^2$ g$^{-1}$) values in section 3.2.

Furthermore, we observed three peaks at 860, 1280–1260, and 1640 cm$^{-1}$, demonstrating the presence

of organic-nitrate (C–ONO$_2$) and oxygenated phenolic groups (Day et al., 2010; Zhang et al., 2020).
Previous studies have shown that the anthropogenic volatile organic compounds, sulfates, nitrates and
other acidic particle components from coal and biomass combustion may enhance the contents of these
functional groups through aqueous-phase formation under high humidity conditions (Gilardoni et al.,




2016; Wang et al., 2019; Zhang et al., 2020). Therefore, the FT-IR spectra indicated that all the BrC
samples from six cities have the contribution of secondary generation. Besides, the abundance of
functional groups at these wavenumbers, especially at 1640 cm$^{-1}$, was higher in CD than that in other
cities. These results might indicate that the secondary source of BrC was relatively high in CD.

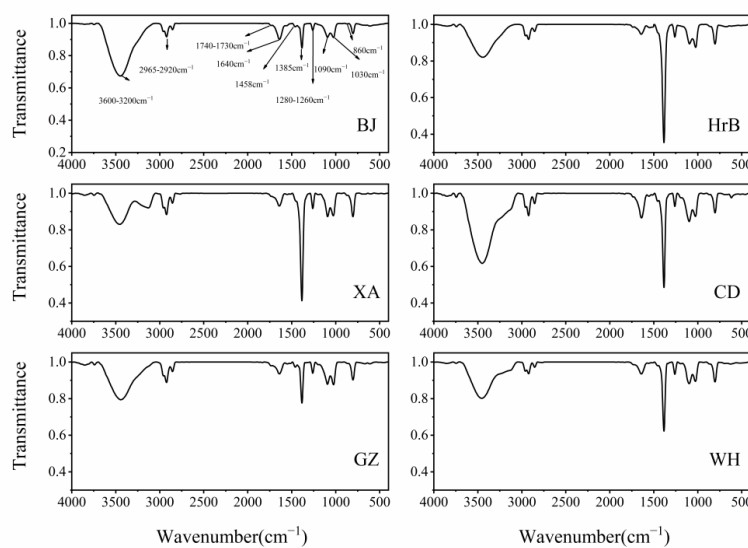

**Figure 4. FTIR spectra of BrC in six megacities.**
**3.4 Source apportionment of BrC**

Considering the complexity of atmospheric processes, and the correlation and/or nonlinear interaction

between independent variables (i.e., multicomponent or multi-source interactions), we attempted to apply
ANN techniques of nonlinear functions, such as MLP model, combined PMF analysis to predict the
source contribution of allocated BrC from PM$_{2.5}$ sources in this study. The PMF-apportioned source
contributions to PM$_{2.5}$ in the six cities are presented in Figures S4 and S5. A strong linear correlation was
observed between the measured and PMF-reconstructed PM$_{2.5}$ mass concentrations ($r$ = 0.90–0.99 in the
six cities), demonstrating the validity and robustness of our PMF solutions. As illustrated in Figure S4,
the first source was dominated by sulfate, OC, and EC and was considered to represent from coal
combustion (Huang et al., 2014). The second source comprised high concentrations of NH$_4^+$, NO$_3^-$, and
SO$_4^{2-}$ and was considered to represent secondary formation processes (Shen et al., 2010). Furthermore,
the third source comprised high loadings of K$^+$ and was considered to represent biomass burning (Shen





et al., 2010). The fourth source primarily comprised Na$^+$, Mg$^{2+}$, and Ca$^{2+}$ and was thus determined to
represent fugitive dust (Shakeri et al., 2016; Shen et al., 2016; Sun et al., 2019). The fifth source contained
high concentrations of Mg$^{2+}$, Ca$^{2+}$, NO$_3^-$, OC, and EC and was thus identified as representing traffic-
related emissions (Shakeri et al., 2016). Finally, the sixth source comprised high concentrations of OC,
EC, and NO$_3^-$ and was considered to represent vehicle emissions (Shakeri et al., 2016).

The optimal neural network model for each site were explored by changing activation function types

(Tan H and Sigmoid), optimizing algorithms (scaled conjugate and gradient descent), and based on the
lowest root mean square error (RMSE) and the highest correlation coefficient (*r*) between observed and
MLP-modelled values (Borlaza et al., 2021). Although there are other architectures that are more
complex for MLP models, a basic MLP architecture was considered sufficient for the input and output
data sets of this study.

Figure S6 shows the correlation between observed values and BrC b$_{abs365}$ predicted values from

selected MLP models. The good correlation indicated the reliability of the model results. On the basis of
the MLP results, we calculated the source-specific contributions to BrC in the six cities (Figure 5). The
primary sources including coal combustion, dust, vehicle, biomass burning and traffic emissions, and
their average contribution to BrC in the northern cities was 93.3%, which was 1.2 times higher than that
in the southern cities. Among these primary emissions, we noted that a higher contribution of biomass
burning to BrC in HrB, BJ, XA and WH compared to other cities, which is consistent with the higher
abundance of biomass burning products, such as aromatic phenol functional groups was founded in these
cities as discussed in section 3.3. As supported, the BrC from biomass burning have high MAE$_{365}$ values
(Cao et al., 2021; Kumar et al., 2018), which can be also observed among these cities (range: 0.68–1.20
m$^2$ g$^{-1}$). On average, the secondary formation source contribution to BrC in southern cities was 19.4%,
which was 2.9 times higher than that in northern cities. Besides, the highest contribution was observed
in CD with 21.2%, followed by GZ > WH > BJ > HrB > XA. This result can be supported by the
abundance of organic-nitrate functional groups, the relatively high AAE value and low MAE$_{365}$ value in
CD, which were closely related with the contribution of secondary sources.



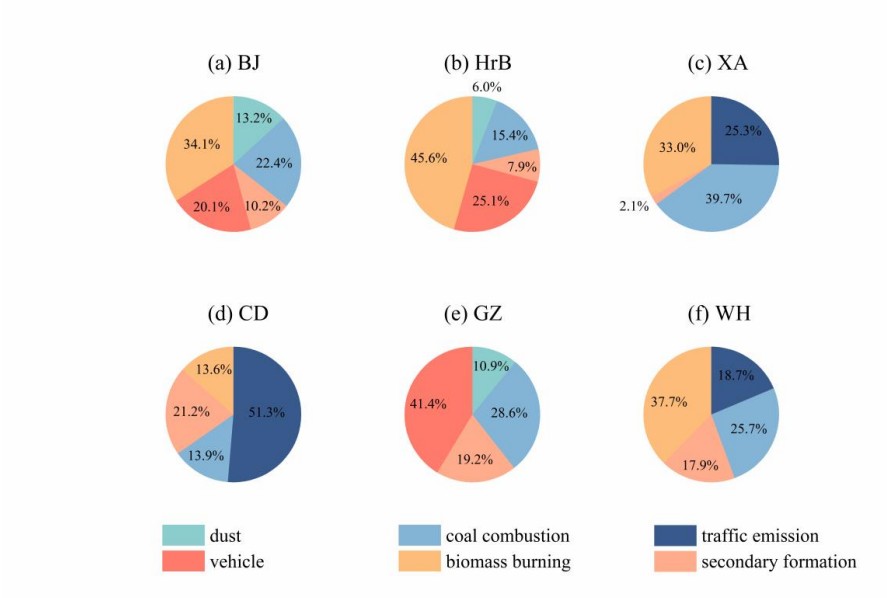

**Figure 5. The source contribution to BrC using multilayer perceptron neural network analysis in (a) BJ, (b) HrB, (c) XA, (d) CD, (e) GZ, (f) WH.**

**4 Conclusions**

We investigated the sources and light absorption properties of BrC in wintertime in six megacities across China. Both the $b_{abs}$ and $MAE_{365}$ of BrC at 365 nm in northern cities were approximately 2.5 and 1.8 times higher than those in southern cities. The $MAE_{365}$ values measured for BrC in BJ, HrB, XA and WH were ranged from 0.68 to 1.21 $m^2$ $g^{-1}$, which were within the MAE ranges derived for biomass burning. Thus, these comparisons confirmed that emissions from biomass burning might be the major BrC source in winter in these cities. Previous studies have reported that $MAE_{365}$ values decreased with aerosol aging while the AAE values of SOA were higher than those for POA. Besides, we noticed that the average $MAE_{365}$ and AAE values showed different trends in southern cities of CD and GZ, that is, the $MAE_{365}$ values of these two cities were lower than those of other cities, while the AAE values were relatively higher. These evidences supported the secondary formation process were among the main sources of BrC in CD and GZ.

The chemical functional groups of BrC in six cities mainly included aliphatic chains, carboxyl groups and aromatic groups. However, the apparent difference of typical functional bands revealed the important



contributions of primary biomass burning and coal combustion to BrC for high abundance of oxygenated
phenolic compounds in these cities, especially in HrB, XA and WH. In contrast, the presence of organic-
nitrate (C–ONO$_2$) and oxygenated phenolic groups in BrC molecular implied the contribution from
secondary formation in six megacities, especially in CD city.
Due to the complexity of atmospheric processes, which are usually non-linear in nature, and the
traditional linear-based source analytic models may be limited. Here, we used a multilayer perceptron
(MLP) model based on artificial neural network (ANN) to improve the source allocation of BrC in these
cities. Source apportionment of BrC based on PMF and ANN-MLP analysis revealed that primary
emissions (e.g., biomass burning, coal combustion, and vehicle emissions) were key contributors to BrC,
and their average contribution in northern cities was about 93.3%, which was 1.2 times higher than that
in southern cities. Secondary formation processes made a greater contribution to BrC in southern cities
(19.4%) than northern cities (6.7%). The results of our work can provide a basis for the development of
more effective practices to control BrC emissions at the regional level.

**Data availability.** The key data sets are publicly available on the Zendo data repository platform:
https://zenodo.org/record/6790321.

**Author contribution.** ZS: Conceptualization. DW, TZ, and SH: Data curation. ZS and HX: Funding
acquisition. DW and YL: Methodology. ZS and JC: Resources. DW: Writing - original draft. DW, ZS,
QZ, HX, JS, JC and YL: Writing - review & editing.

**Competing interests.** The authors declare that they have no conflict of interest.

**Acknowledgements.** This research was supported by the National Natural Science Foundation of China
(41877383) and State Key Laboratory of Loess and Quaternary Geology, Institute of Earth Environment,
CAS (SKLLQG2103). Authors also thanks for Dr. Jun Tao, Renjian Zhang, Shaofei Kong, and Song Cui
for their help in field sampling.

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
