# Peer review of "Winter brown carbon over six China's megacities: Light"

_EGUsphere, 2022_

## Referee Comment (RC2)

General Comments

This study reports the absorption properties, chemical functional bonds, and sources of BrC in six Chinese cities. They were conducted to acquire comprehensive BrC datasets with higher resolution and more species and provide insight into more specified source identification. Overall, the combination and intercomparison of BrC in six Chinese cities is a valuable study of the BrC measurement. I recommend publication after the following issues are addressed.

P7 Line 188

As far as I know, "Elser et al., 2016" reported the AMS data in China, the result is about BBOA is the major source for OA in Xi'an, it was not included any results for the brown carbon. Please revise it.

P11 Line 265-280

In Figure 4, there are the big differences between cities at the peak of 1385 cm-1, why? What are the insights on the difference?

P12 Line 292-308

My concern is related to the results of PMF. PMF is a commonly used receptor model for source apportionment. From the shown factor profiles in Figure S4. The results of PMF are highly associated/influenced by the sample number (not many here), chemical components (could be enlarged here), and the suggested factors (4, 5, 6, or even more). There are only 30 (or 31?) samples in each city and different profiles in each city. It could be more careful. And here needs more evidence.

Also, there are some studies about the source apportionment of BrC, which directly input the MAE365 of BrC to the data matrix. Did the author try this solution? How about the results?

---

## Author Comment (AC1)

General Comments

This study reports the absorption properties, chemical functional bonds, and sources of BrC in six Chinese cities. They were conducted to acquire comprehensive BrC datasets with higher resolution and more species and provide insight into more specified source identification. Overall, the combination and intercomparison of BrC in six Chinese cities is a valuable study of the BrC measurement. I recommend publication after the following issues are addressed.

**Response to the Editor and Reviewer**

We greatly appreciate editor and anonymous reviewer for their constructive comments and suggestions, which greatly assist to improve the quality of our manuscript. We have responded to all of the points accordingly. The original comments are in black and our responses are in blue. Major changes are highlighted in the revised paper. We hope you and the reviewers will find the revised version meets the standard of the journal.

P7 Line 188

As far as I know, "Elser et al., 2016" reported the AMS data in China, the result is about BBOA is the major source for OA in Xi'an, it was not included any results for the brown carbon. Please revise it.

**Response:** Suggestion taken. The sentence was revised as:

Lines 200-202:

Biomass burning was revealed to be the dominant source of BrC in these cities during winter (Cheng et al., 2016; Shen et al., 2017; Sun et al., 2017; Cheng et al., 2022).

Reference:

Cheng, Y., Cao, X. B., Liu, J. M., Yu, Q. Q., Wang, P., Yan, C. Q., Du, Z. Y., Liang, L. L., Zhang, Q., and He, K. B.: Primary nature of brown carbon absorption in a frigid atmosphere with strong haze chemistry, Environ Res, 204, 112324, https://doi.org/10.1016/j.envres.2021.112324, 2022.

Cheng, Y., He, K. B., Du, Z. Y., Engling, G., Liu, J. M., Ma, Y. L., Zheng, M., and Weber, R. J.: The characteristics of brown carbon aerosol during winter in Beijing, Atmos. Environ., 127, 355-364, https://doi.org/10.1016/j.atmosenv.2015.12.035, 2016.

Shen, Z., Zhang, Q., Cao, J., Zhang, L., Lei, Y., Huang, Y., Huang, R. J., Gao, J., Zhao, Z., Zhu, C., Yin, X., Zheng, C., Xu, H., and Liu, S.: Optical properties and possible sources of brown carbon in $PM_{2.5}$ over Xi'an, China, Atmos. Environ., 150, 322-330, http://dx.doi.org/10.1016/j.atmosenv.2016.11.024, 2017.

Sun, J., Shen, Z., Cao, J., Zhang, L., Wu, T., Zhang, Q., Yin, X., Lei, Y., Huang, Y., Huang, R. J., Liu, S., Han, Y., Xu, H., Zheng, C., and Liu, P.: Particulate matters emitted from maize straw burning for winter heating in rural areas in Guanzhong Plain, China: Current emission and future reduction, Atmos. Res., 184, 66-76, http://dx.doi.org/10.1016/j.atmosres.2016.10.006, 2017.

P11 Line 265-280

In Figure 4, there are the big differences between cities at the peak of 1385 cm-1, why?

What are the insights on the difference?

**Response:** Previous studies have found a sharp peak near 1385 cm$^{-1}$ with particularly high intensity in the FT-IR spectra of atmospheric BrC samples, which may be related to O−H bond deformation and C−O stretching of phenolic groups (Mukherjee et al., 2020; Zhang et al., 2020). In this study, we found the similar peak at 1385cm$^{-1}$, thus we inferred that the same phenolic functional groups may also exist in the BrC samples in our study. In addition, previous researches have pointed out the presence of peak at 1385 cm$^{-1}$, which suggests that the contribution of biomass burning source to BrC. Since fine smoke aerosol contain thermally altered lignin derivatives, such as aromatic phenols (e.g., siringyl and guaiacyl derivatives) (Duarte et al., 2007; Mukherjee et al., 2020). The same peak was observed in the FT-IR spectra of BrC samples derived from the combustion of biomass materials (Fan et al., 2016). In our study, we noted that the abundance at 1385 cm$^{-1}$ varies greatly among six cities, which was significantly higher in HrB, XA and WH than other cities. We speculated that this is due to the different contribution of biomass burning among cities, and the higher contribution occur in HrB, XA and WH. Therefore, we revised these sentences in the manuscript as following:

Lines 285-293:
Moreover, the peak at 1385 cm$^{-1}$ was generally considered to be derived from the O–H bond deformation and C–O stretching of phenolic groups (Fan et al., 2016; Mukherjee et al., 2020; Zhang et al., 2020), and the same peak was observed in the FT-IR spectra of BrC samples derived from the combustion of biomass materials (Fan et al., 2016). These observations indicated the contribution of biomass burning to BrC; this was because that biomass burning can release heat-modified lignin derivatives such as aromatic phenols (e.g., syringyl and guaiacyl) (Duarte et al., 2007; Fan et al., 2016; Zhao et al., 2022). It was noted that the abundance of this peak was different among six cities, and was significantly higher in HrB, XA and WH, which indicated biomass burning contributed differently to BrC in six cities, and higher contribution was occur in HrB, XA and WH than those in other cities.

Reference:

Duarte, R. M. B. O., Santos, E. B. H., Pio, C. A., and Duarte, A. C.: Comparison of structural features of water-soluble organic matter from atmospheric aerosols with those of aquatic humic substances, Atmos. Environ., 41, 8100-8113, https://doi.org/10.1016/j.atmosenv.2007.06.034, 2007.
Fan, X., Wei, S., Zhu, M., Song, J., and Peng, P. a.: Comprehensive characterization of humic-like substances in smoke PM$_{2.5}$ emitted from the combustion of biomass materials and fossil fuels, Atmos. Chem. Phys., 16, 13321-13340, https://doi.org/10.5194/acp-16-13321-2016, 2016.
Mukherjee, A., Dey, S., Rana, A., Jia, S., Banerjee, S., and Sarkar, S.: Sources and atmospheric processing of brown carbon and HULIS in the Indo-Gangetic Plain: Insights from compositional analysis, Environ. Pollut., 267, 115440, https://doi.org/10.1016/j.envpol.2020.115440, 2020.
Zhang, Q., Shen, Z., Zhang, L., Zeng, Y., Ning, Z., Zhang, T., Lei, Y., Wang, Q., Li, G., Sun, J., Westerdahl, D., Xu, H., and Cao, J.: Investigation of primary and secondary particulate brown

carbon in two Chinese cities of Xi'an and Hong Kong in wintertime, Environ. Sci. Technol., 54, 3803-3813, https://dx.doi.org/10.1021/acs.est.9b05332, 2020.

Zhao, R., Zhang, Q., Xu, X., Wang, W., Zhao, W., Zhang, W., and Zhang, Y.: Light absorption properties and molecular compositions of water-soluble and methanol-soluble organic carbon emitted from wood pyrolysis and combustion, Sci. Total. Environ., 809, 151136, https://doi.org/10.1016/j.scitotenv.2021.151136, 2022.

P12 Line 292-308

My concern is related to the results of PMF. PMF is a commonly used receptor model for source apportionment. From the shown factor profiles in Figure S4. The results of PMF are highly associated/influenced by the sample number (not many here), chemical components (could be enlarged here), and the suggested factors (4, 5, 6, or even more). There are only 30 (or 31?) samples in each city and different profiles in each city. It could be more careful. And here needs more evidence.

**Response:** Suggestion taken. In order to extract four to six factors, the PMF of six cities were run multiple times during building of PMF models. Solutions with fewer or greater number factors were also investigated, but these solutions were less defined and factor merging was often observed. Finally, according to the highest correlation coefficient between the constructed $PM_{2.5}$ concentration and the measured $PM_{2.5}$ concentration, the characteristic factors used for source identification were consistent with the literature to determine the optimal model for each site, and thus to reduce the influence of the limited number of samples and chemical species on the PMF results. we provided additional evidences to show the plausibility of the PMF model results:

Line 314-319:

A good correlation was observed between the measured and PMF-reconstructed $PM_{2.5}$ mass concentrations in all sites (BJ: r = 0.99; HrB: r = 0.90; XA: r = 0.97; CD: r = 0.97; GZ: r = 0.94; WH: r = 0.95), theoretical $Q_{true}$ and $Q_{robust}$ displayed a <5% difference, and the scaled residuals for final model solutions were generally normally distributed, falling into the recommended range of −3 to 3. These evidences demonstrating the validity and robustness of our PMF solutions (Borlaza et al., 2021b; Tao et al., 2021).

Reference:
Borlaza, L. J. S., Weber, S., Uzu, G., Jacob, V., Cañete, T., Micallef, S., Trébuchon, C., Slama, R., Favez, O., and Jaffrezo, J.-L.: Disparities in particulate matter ($PM_{10}$) origins and oxidative potential at a city scale (Grenoble, France) – Part 1: Source apportionment at three neighbouring sites, Atmos. Chem. and Phys., 21, 5415-5437, https://doi.org/10.5194/acp-21-5415-2021, 2021b.

Tao, Y., Sun, N., Li, X., Zhao, Z., Ma, S., Huang, H., Ye, Z., and Ge, X.: Chemical and Optical Characteristics and Sources of $PM_{2.5}$ Humic-Like Substances at Industrial and Suburban Sites in Changzhou, China, Atmosphere, 12, https://doi.org/10.3390/atmos12020276, 2021.

Also, there are some studies about the source apportionment of BrC, which directly input the MAE365 of BrC to the data matrix. Did the author try this solution? How about the results?

**Response:** We thank the reviewer's suggestion. We tried to run the PMF model by putting BrC $b_{abs365}$ into the data matrix to get the BrC source resolution results directly. However, the correlation between the measured value and the constructed value of $b_{abs365}$ was low, and there was a great difference between the theoretical $Q_{true}$ and $Q_{robust}$, which indicated that the PMF model results have great uncertainty and low credibility (Borlaza et al., 2021a; Tao et al., 2021). We speculated that this may be caused by the fact that our samples are not suitable for the PMF model. As a bilinear factor analysis method, PMF is widely used for source assignment (Cao et al., 2012; Lei et al., 2018; Li et al., 2021; Shen et al., 2010; Tao et al., 2017). However, atmospheric processes are nonlinear in nature, the interaction between sources cannot be ignored (Borlaza et al., 2021b), which may be the reason why this linear fitting is not suitable for our study. The artificial neural network (ANN) based multilayer perceptron (MLP) model could extract trends from non-linear data, making it an interesting and competitive innovative analytical approach for many scientific disciplines (Borlaza et al., 2021b; Elangasinghe et al., 2014). Therefore, we decided to use a combination of PMF and ANN-MLP model to assign source contributions to BrC.

Reference:
Borlaza, L. J. S., Weber, S., Uzu, G., Jacob, V., Cañete, T., Micallef, S., Trébuchon, C., Slama, R., Favez, O., and Jaffrezo, J.-L.: Disparities in particulate matter ($PM_{10}$) origins and oxidative potential at a city scale (Grenoble, France) – Part 1: Source apportionment at three neighbouring sites, Atmos. Chem. and Phys., 21, 5415-5437,
https://doi.org/10.5194/acp-21-5415-2021, 2021a.
Borlaza, L. J. S., Weber, S., Jaffrezo, J.-L., Houdier, S., Slama, R., Rieux, C., Albinet, A., Micallef, S., Trébluchon, C., and Uzu, G.: Disparities in particulate matter ($PM_{10}$) origins and oxidative potential at a city scale (Grenoble, France) – Part 2: Sources of $PM_{10}$ oxidative potential using multiple linear regression analysis and the predictive applicability of multilayer perceptron neural network analysis, Atmos. Chem. Phys., 21, 9719-9739, https://doi.org/10.5194/acp-21-9719-2021, 2021b.
Cao, J. J., Wang, Q. Y., Chow, J. C., Watson, J. G., Tie, X. X., Shen, Z. X., Wang, P., and An, Z. S.: Impacts of aerosol compositions on visibility impairment in Xi'an, China, Atmos. Environ., 59, 559-566, https://doi.org/10.1016/j.atmosenv.2012.05.036, 2012.
Elangasinghe, M. A., Singhal, N., Dirks, K. N., and Salmond, J. A.: Development of an ANN–based air pollution forecasting system with explicit knowledge through sensitivity analysis, Atmos. Pollut. Res., 5, 696-708, https://doi.org/10.5094/APR.2014.079, 2014.
Lei, Y., Shen, Z., Wang, Q., Zhang, T., Cao, J., Sun, J., Zhang, Q., Wang, L., Xu, H., Tian, J., and Wu, J.: Optical characteristics and source apportionment of brown carbon in winter $PM_{2.5}$ over Yulin in Northern China, Atmos. Res., 213, 27-33, https://doi.org/10.1016/j.atmosres.2018.05.018, 2018.
Li, X., Zhao, Q., Yang, Y., Zhao, Z., Liu, Z., Wen, T., Hu, B., Wang, Y., Wang, L., and Wang, G.: Composition and sources of brown carbon aerosols in megacity Beijing during the winter of 2016,

Atmos. Res., 262, https://doi.org/10.1016/j.atmosres.2021.105773, 2021.

Shen, Z., Cao, J., Arimoto, R., Han, Y., Zhu, C., Tian, J., and Liu, S.: Chemical Characteristics of Fine Particles ($PM_1$) from Xi'an, China, Aerosol. Sci. Tech., 44, 461-472, https://doi.org/10.1080/02786821003738908, 2010.

Tao, J., Zhang, L., Cao, J., Zhong, L., Chen, D., Yang, Y., Chen, D., Chen, L., Zhang, Z., Wu, Y., Xia, Y., Ye, S., and Zhang, R.: Source apportionment of $PM_{2.5}$ at urban and suburban areas of the Pearl River Delta region, south China - With emphasis on ship emissions, Sci. Total. Environ., 574, 1559-1570, http://dx.doi.org/10.1016/j.scitotenv.2016.08.175, 2017.

---

## Author Comment (AC2)

This manuscript described the light absorption, molecular characterization, and source apportionment of winter brown carbon over six China's megacities. The BrC light absorption coefficient and the mass absorption efficiency at 365 nm in northern cities were higher than those in southern cities by 2.5 and 1.8 times, respectively. The new method of Positive matrix factorization (PMF) coupled with multilayer perceptron (MLP) neural network analysis were used to apportion the sources of BrC light absorption. The results highlighted that primary emissions made a major contribution to BrC in six megacities. While secondary formation processes made a greater contribution to light absorption in the southern cities. The following comments should be considered to improve the manuscript.

**Response to the Editor and Reviewer**

We greatly appreciate the editor and the referee for their careful and constructive comments, which have helped us improve the paper quality significantly. We have addressed all of the comments raised by the reviewers. Our point-by-point responses are detailed below. The original comments are in black and our responses are in blue. Major changes are highlighted in the revised paper. We hope you and the reviewers will find the revised version meets the standard of the journal.

1\ Line 42-44: "BrC… including incomplete combustion of fossil fuels, biomass burning, forest fires, and residential coal combustion (Kirchstetter et al., 2004; Shen et al., 2017; Soleimanian et al., 2020).". Avoid lumping references.

**Response:** Suggestion taken. The sentence was revised as:
Lines 43-45:
BrC in urban atmospheres can originate from numerous sources, including incomplete combustion of fossil fuels (Soleimanian et al., 2020), biomass burning (Shen et al., 2017; Soleimanian et al., 2020), forest fires, and residential coal combustion (Kirchstetter et al., 2004; Soleimanian et al., 2020).

2\ Line 53: Avoid lumping references as in (Cheng et al., 2016; Kim et al., 2016; Mo et al., 2021; Shen et al., 2017). Instead summarize the main contribution of each referenced paper in a separate sentence. For example, "Furthermore, … in Beijing (Cheng et al., 2016), Xi'an (Kim et al., 2016) …biomass burning emissions.".

**Response:** Suggestion taken. The sentence was revised as:
Lines 54-57:
Furthermore, a stronger light absorption ability in cold seasons (fall and winter) in Beijing (Cheng et al., 2016), Xi'an (Shen et al., 2017), Seoul (Kim et al., 2016), Taiyuan and other cities (Mo et al., 2021) has been found to be strongly associated with increased biomass burning emissions.

3\ Line 63-64: Please briefly describe the results for BrC source elucidation in these literatures after this sentence.

**Response:** Suggestion taken. We add the description of the results from these literatures in the revised manuscript.

Lines 72-79:

For example, Bao et al. (2022) obtained specific source contributions to BrC $b_{abs365}$ in Nanjing based on PMF and MLR method, confirming that the key contributors to BrC $b_{abs365}$ were mainly derived from biomass burning, primary industrial and traffic emissions. Lei et al. (2018) investigated the source apportionment of BrC $b_{abs365}$ in Yulin and showed that the residential coal combustion was the highest contributor to BrC $b_{abs365}$ in winter. Soleimanian et al. (2020) used the principal component analysis (PCA) coupled with MLR source apportionment model, which identified fossil fuel combustion was the dominant source of BrC $b_{abs365}$ in central Los Angeles during summer (38%), followed by SOA (30%) and biomass burning (12%).

4\ More references on BrC should be involved in the Introduction.

**Response:** Suggestion taken. We have added the content of BrC research status.
Lines 52-54:
Mo et al (2021) studied the light absorption coefficient of BrC at 365 nm (BrC $b_{abs365}$) in ten Chinese cities, which found that the BrC $b_{abs365}$ value displayed obvious spatial (northern China > southern China) variations.
Lines 57-61:
The mass absorption efficiency at 365 nm (MAE$_{365}$) of BrC has been widely used to evaluate the light-absorbing ability of BrC (Bao et al., 2022). Xie et al (2017) found that the BrC MAE$_{365}$ values from biomass burning ($1.28 \pm 0.12$ m$^2$ g$^{-1}$) were higher than those from vehicle emissions ($0.62 \pm 0.76$ m$^2$ g$^{-1}$). Ni et al (2021) noted that BrC MAE$_{365}$ values can be decreased from 1.43 m$^2$ g$^{-1}$ to 0.11 m$^2$ g$^{-1}$ with the BrC aerosol aged.

Reference:
Bao, M., Zhang, Y. L., Cao, F., Lin, Y. C., Hong, Y., Fan, M., Zhang, Y., Yang, X., and Xie, F.: Light absorption and source apportionment of water soluble humic-like substances (HULIS) in PM$_{2.5}$ at Nanjing, China, Environ. Res., 206, 112554, https://doi.org/10.1016/j.envres.2021.112554, 2022.
Mo, Y., Li, J., Cheng, Z., Zhong, G., Zhu, S., Tian, C., Chen, Y., and Zhang, G.: Dual carbon isotope-based source apportionment and light absorption properties of water-soluble organic carbon in PM$_{2.5}$ over China, J. Geophys. Res. Atmos., 126, https://doi.org/10.1029/2020JD033920, 2021.
Ni, H., Huang, R. J., Pieber, S. M., Corbin, J. C., Stefenelli, G., Pospisilova, V., Klein, F., Gysel-Beer, M., Yang, L., Baltensperger, U., Haddad, I. E., Slowik, J. G., Cao, J., Prevot, A. S. H., and Dusek, U.: Brown carbon in primary and aged coal combustion emission, Environ. Sci. Technol., 55, 5701-5710,
https://doi.org/10.1021/acs.est.0c08084, 2021.
Xie, M., Hays, M. D., and Holder, A. L.: Light-absorbing organic carbon from prescribed and laboratory biomass burning and gasoline vehicle emissions, Sci. Rep., 7, 7318,

https://doi.org/10.1038/s41598-017-06981-8, 2017.

5\ Line 83-94: Please supplement the information about the sampling location, such as what is the main impact on the surrounding environment.

**Response:** As described in section 2.1: "Information about the six cities and the sampling sites is summarized in Table S1", detailed information about the sampling site, such as sampling height, latitude and longitude, and major emission sources in the surrounding environment has been described in the Table S1 in Supplementary information.

**Table S1. Information of sampling sites**

| Observation megacity | Location | Geographical China | Site description |
|---|---|---|---|
| Beijing | 39.97° N, 116.36° E | North China | ~8 m above ground level, in the north part of Beijing, which is close to several major roads including a highway and is surrounded by residences and restaurants. |
| Harbin | 45.74° N, 126.73° E | | ~18 m above ground level, in the east of Harbin, surrounded by campus, roads, residential commercial emission sources. |
| Xi'an | 34.23° N, 108.88° E | | ~15 m above ground level, in the southeast of downtown Xi'an, surrounded by two lane roads, residential commercial districts. |
| Chengdu | 30.70° N, 104.06° E | | ~18 m above ground level, on the rooftop of a building of Southwest Jiaotong University, surrounded by commercial and residential areas and close to a train station. |
| Guangzhou | 23.12° N, 113.35° E | South China | ~30 m above ground level, in the central of Guangzhou, there is no obvious industrial pollution source near the monitoring station. |
| Wuhan | 30.53° N, 114.39° E | | ~18 m above ground level, in the southeast of Wuhan city, surrounded by roads, residential commercial districts, this is a typical urban site with no industrial emission sources nearby. |

6\ Line 123: Please explain why the light absorption coefficient at 365nm was chosen as a representation of the light absorption characteristics of the BrC.

**Response:** The light absorption coefficient at 365 nm was chosen to represent BrC light absorption characteristics has been widely used in other studies (Bao et al., 2022; Cheng et al., 2016; Mo et al., 2021). The wavelength of 365 nm was chosen to avoid interference of light absorption of non-organic compounds (e.g., nitrate) (Cheng et al., 2016; Huang et al., 2018).

Reference:

Bao, M., Zhang, Y. L., Cao, F., Lin, Y. C., Hong, Y., Fan, M., Zhang, Y., Yang, X., and Xie, F.: Light absorption and source apportionment of water soluble humic-like substances (HULIS) in $PM_{2.5}$ at Nanjing, China, Environ. Res., 206, 112554, https://doi.org/10.1016/j.envres.2021.112554, 2022.

Cheng, Y., He, K. B., Du, Z. Y., Engling, G., Liu, J. M., Ma, Y. L., Zheng, M., and Weber, R. J.: The characteristics of brown carbon aerosol during winter in Beijing, Atmos. Environ., 127, 355-364, https://doi.org/10.1016/j.atmosenv.2015.12.035, 2016.

Huang, R. J., Yang, L., Cao, J., Chen, Y., Chen, Q., Li, Y., Duan, J., Zhu, C., Dai, W., Wang, K., Lin,

C., Ni, H., Corbin, J. C., Wu, Y., Zhang, R., Tie, X., Hoffmann, T., O'Dowd, C., and Dusek, U.: Brown Carbon Aerosol in Urban Xi'an, Northwest China: The composition and light absorption properties, Environ. Sci. Technol., 52, 6825-6833, https://doi.org/10.1021/acs.est.8b02386, 2018.

Mo, Y., Li, J., Cheng, Z., Zhong, G., Zhu, S., Tian, C., Chen, Y., and Zhang, G.: Dual carbon isotope-based source apportionment and light absorption properties of water-soluble organic carbon in $PM_{2.5}$ over China, J. Geophys. Res. Atmos., 126, https://doi.org/10.1029/2020JD033920, 2021.

7\ Please unify to unit format, such as $m^2\ g^{-1}$ and $\mu g/m^3$.

**Response:** Suggestion taken. The unit format $\mu g/m^3$ is uniformly modified to $\mu g\ m^{-3}$.

8\ Line 154-156: "… this can primarily be attributed to substantial emissions from residential heating in winter in northern China." According to the above-mentioned average OC concentration in northern cities is higher than that in southern cities, it seems difficult to infer that this is caused by heating in northern China in winter. A more logical explanation should be that the burning of fuels used for winter heating in northern cities, such as coal, produces high OC values, resulting in higher OC values in northern cities than southern cities. Please make corresponding modifications.

**Response:** Suggestion taken. The sentence was revised as:
Lines 171-175:
Similar to the $PM_{2.5}$ trend, the average OC concentration in the northern cities ($15.5 \pm 7.9\ \mu g\ m^{-3}$) was higher than that in the southern cities ($9.2 \pm 4.6\ \mu g\ m^{-3}$), which can be attributed to substantial emissions from residential heating (i.e., coal and biomass combustion) in winter in northern China (Zhang et al., 2021). In addition, these residential fuels can emit an abundant OC emission (Lei et al., 2018; Sun et al., 2017).

9\ Line 157-158: Please add a description of the method for calculating POC and SOC using the EC tracer method in the Supplementary information.

**Response:** Suggestion taken. The method for calculating POC and SOC using EC tracer method has been added in text S2 in the Supplementary information, and the sentence has been revised as:
Line 177:
Detailed calculation method was described in Text S3.
Lines 83-88 in Supplementary information:
Text S3. SOC and POC calculation.
   To assess the sources of atmospheric BrC, primary OC (POC) and secondary OC (SOC) were estimated by using the EC tracer method (Ram and Sarin, 2011) as in the following equation:

$$SOC = OC_{tot}\text{-}EC \times (OC/EC)_{min} \tag{S5}$$

$$POC= OC_{tot} \text{-}SOC \tag{S6}$$

where $OC_{tot}$ is total OC, $(OC/EC)_{min}$ is the minimum OC/EC ratio observed at each site.

10\ Line 191: "The aging or oxidation of aerosols was confirmed to be the major…" The word "was" should be "were".

**Response:** Suggestion taken.

11\ Fig 2. As I can see, the $MAE_{365}$ is higher in Wuhan, can you give some explanation?

**Response:** As described in our article, the BrC $MAE_{365}$ values are in the following order across the six cities: HrB > BJ > WH > XA > GZ > CD. Therefore, the highest BrC $MAE_{365}$ can be observed in WH among southern cities. As shown in Section 3.4 of our study, the contribution of biomass combustion to BrC in WH (37.7%) could not be ignored, which is higher than that in Chengdu (13.6%) and Guangzhou (0%). As supported, the BrC derived from biomass burning usually has a high MAE value (Xie et al., 2017; Soleimanian et al., 2020; Zhang et al., 2021b). Therefore, the high emission contribution of biomass combustion sources was the reason for the high $MAE_{365}$ value in WH.
We revised the manuscript in Lines 345-347:
In addition, we noted that the contribution of biomass burning to BrC in WH (37.7%) was higher than that in CD (13.6%) and GZ (0%), which can explain the highest BrC $MAE_{365}$ was observed in WH among southern cities as shown in Figure 2.

Reference:
Soleimanian, E., Mousavi, A., Taghvaee, S., Shafer, M. M., and Sioutas, C.: Impact of secondary and primary particulate matter (PM) sources on the enhanced light absorption by brown carbon (BrC) particles in central Los Angeles, Sci. Total. Environ., 705, https://doi.org/10.1016/j.scitotenv.2019.135902, 2020.
Xie, M., Hays, M. D., and Holder, A. L.: Light-absorbing organic carbon from prescribed and laboratory biomass burning and gasoline vehicle emissions, Sci. Rep., 7, 7318, https://doi.org/10.1038/s41598-017-06981-8, 2017.
Xiong, Y., Zhou, J., Schauer, J. J., Yu, W., and Hu, Y.: Seasonal and spatial differences in source contributions to PM2.5 in Wuhan, China, Sci. Total. Environ., http://dx.doi.org/10.1016/j.scitotenv.2016.10.150, 2016.
Zhang, Q., Shen, Z., Zhang, T., Kong, S., Lei, Y., Wang, Q., Tao, J., Zhang, R., Wei, P., Wei, C., Cui, S., Cheng, T., Ho, S. S. H., Li, Z., Xu, H., and Cao, J.: Spatial distribution and sources of winter black carbon and brown carbon in six Chinese megacities, Sci. Total. Environ., 762, 143075, https://doi.org/10.1016/j.scitotenv.2020.143075, 2021a.
Zhang, T., Shen, Z., Zeng, Y., Cheng, C., Wang, D., Zhang, Q., Lei, Y., Zhang, Y., Sun, J., Xu, H., Ho, S. S. H., and Cao, J.: Light absorption properties and molecular profiles of HULIS in PM2.5 emitted from biomass burning in traditional "Heated Kang" in Northwest China, Sci. Total. Environ., https://doi.org/10.1016/j.scitotenv.2021.146014, 2021b.

12\ What is the difference between the light absorption coefficient ($b_{abs365}$) and the mass absorption efficiency ($MAE_{365}$)?

**Response:** $b_{abs365}$ represents the light absorption coefficient at wavelength of 365 nm for BrC (Huang et al., 2018). $MAE_{365}$ is usually obtained from the $b_{abs365}$/OC, and refers to the light absorption capacity of per unit BrC mass (Cheng et al., 2016; Shen et al., 2017).

Reference:
Cheng, Y., He, K. B., Du, Z. Y., Engling, G., Liu, J. M., Ma, Y. L., Zheng, M., and Weber, R. J.: The characteristics of brown carbon aerosol during winter in Beijing, Atmos. Environ., 127, 355-364, https://doi.org/10.1016/j.atmosenv.2015.12.035, 2016.
Huang, R. J., Yang, L., Cao, J., Chen, Y., Chen, Q., Li, Y., Duan, J., Zhu, C., Dai, W., Wang, K., Lin, C., Ni, H., Corbin, J. C., Wu, Y., Zhang, R., Tie, X., Hoffmann, T., O'Dowd, C., and Dusek, U.: Brown Carbon Aerosol in Urban Xi'an, Northwest China: The composition and light absorption properties, Environ. Sci. Technol., 52, 6825-6833, https://doi.org/10.1021/acs.est.8b02386, 2018.
Shen, Z., Zhang, Q., Cao, J., Zhang, L., Lei, Y., Huang, Y., Huang, R. J., Gao, J., Zhao, Z., Zhu, C., Yin, X., Zheng, C., Xu, H., and Liu, S.: Optical properties and possible sources of brown carbon in $PM_{2.5}$ over Xi'an, China, Atmos. Environ., 150, 322-330,
http://dx.doi.org/10.1016/j.atmosenv.2016.11.024, 2017.

13\ Line 226-228: "…at the Nepal Climate Observatory-Pyramid (3.7–4.0; Kirillova et al., 2016) and in the Los Angeles Basin (4.82 ± 0.49; Zhang et al., 2013) …Tibetan Plateau (8.2 ± 1.4; Zhu et al., 2018) …". The AAE values should be compared in a similar wavelength range. Please add the wavelength range of fitted AAE values in all references cited here.

**Response:** Suggestion taken. The sentence was revised as:
Lines 243-250:
In general, the AAE values obtained in this study are higher than those obtained at the Nepal Climate Observatory-Pyramid (3.7–4.0; 330–500 nm) (Kirillova et al., 2016) and in the Los Angeles Basin (4.82 ± 0.49; 300–600 nm) (Zhang et al., 2013) and lower than those obtained at the Tibetan Plateau (8.2 ± 1.4; 365–550 nm) (Zhu et al., 2018). Nevertheless, the values obtained in this study are comparable to those obtained in Beijing (5.3–7.3; 310–450 nm) (Cheng et al., 2016; Wu et al., 2021), Nanjing (6.7; 300–600 nm) (Chen et al., 2018), the Indo-Gangetic Plain (5.3; 300–700 nm) (Srinivas et al., 2016), New Delhi (5.1; 330–400 nm) (Kirillova et al., 2014), Seoul (5.5–5.8; 300–700 nm) (Kim et al., 2016), and Xi'an (5.3–6.1; 330–550 nm) (Huang et al., 2018).

14\ Line 234-236: "… studies for coal combustion (5.5–6.4; Ni et al., 2021), biomass burning (4.4–8.7; Xie et al., 2017), and gasoline vehicle emissions (6.2–6.9; Xie et al., 2017)." Please add the wavelength range of fitted AAE values in all references cited here.

**Response:** Suggestion taken. The sentence was revised as:

Lines 253-256:

Furthermore, the AAE values obtained in this study are within the range of those reported by previous studies for coal combustion (5.5–6.4; 300–500nm) (Ni et al., 2021), biomass burning (4.4–8.7; 300–550nm) (Xie et al., 2017), and gasoline vehicle emissions (6.2–6.9; 300–550 nm) (Xie et al., 2017).